# Comparison between Neutralization Capacity of Antibodies Elicited by COVID-19 Natural Infection and Vaccination in Indonesia: A Prospective Cohort

**DOI:** 10.3390/antib12030060

**Published:** 2023-09-21

**Authors:** Sitti Nurisyah, Mitsuhiro Iyori, Ammar Abdurrahman Hasyim, Akihiko Sakamoto, Hinata Hashimoto, Kyouhei Yamagata, Saya Yamauchi, Khaeriah Amru, Kartika Hardianti Zainal, Irfan Idris, Shigeto Yoshida, Irawaty Djaharuddin, Din Syafruddin, Agussalim Bukhari, Puji Budi Setia Asih, Yenni Yusuf

**Affiliations:** 1Faculty of Medicine, Hasanuddin University, Makassar 90245, Indonesia; ichanurisyah@gmail.com (S.N.); khaeriahamru@gmail.com (K.A.); irfanfaal@gmail.com (I.I.); irawatymuzakkir@gmail.com (I.D.); dinkarim@yahoo.com (D.S.); agussalim.bukhari@med.unhas.ac.id (A.B.); 2Dr. Tadjuddin Chalid Hospital, Makassar 90241, Indonesia; 3Research Institute of Pharmaceutical Sciences, Musashino University, Nishitokyo, Tokyo 202-8585, Japan; m-iyori@musashino-u.ac.jp; 4Laboratory of Vaccinology and Applied Immunology, Kanazawa University, Kanazawa 920-1192, Japan; ammarhasyim26@gmail.com (A.A.H.); akisaka@p.kanazawa-u.ac.jp (A.S.); hinatabocco1123@stu.kanazawa-u.ac.jp (H.H.); aron@stu.kanazawa-u.ac.jp (K.Y.); ysaya@stu.kanazawa-u.ac.jp (S.Y.); kartikahardianti@gmail.com (K.H.Z.); shigeto@p.kanazawa-u.ac.jp (S.Y.); 5Hasanuddin University Medical Research Centre, Makassar 90245, Indonesia; 6Dr. Wahidin Soedirohusodo Hospital, Makassar 90425, Indonesia; 7Badan Riset dan Inovasi Nasional (National Research and Innovation Agency), Jakarta 10340, Indonesia; puji_bsa@yahoo.com

**Keywords:** COVID-19, receptor binding domain (RBD), natural infection, vaccination, neutralizing antibodies

## Abstract

Background: To fight the COVID-19 pandemic, immunity against SARS-CoV-2 should be achieved not only through natural infection but also by vaccination. The effect of COVID-19 vaccination on previously infected persons is debatable. Methods: A prospective cohort was undergone to collect sera from unvaccinated survivors and vaccinated persons—with and without COVID-19 pre-infection. The sera were analyzed for the anti-receptor binding domain (RBD) titers by ELISA and for the capacity to neutralize the pseudovirus of the Wuhan-Hu-1 strain by luciferase assays. Results: Neither the antibody titers nor the neutralization capacity was significantly different between the three groups. However, the correlation between the antibody titers and the percentage of viral neutralization derived from sera of unvaccinated survivors was higher than that from vaccinated persons with pre-infection and vaccinated naïve individuals (Spearman correlation coefficient (*r*) = −0.8558; 95% CI, −0.9259 to −0.7288), *p* < 0.0001 vs. −0.7855; 95% CI, −0.8877 to −0.6096, *p* < 0.0001 and −0.581; 95% CI, −0.7679 to −0.3028, *p* = 0.0002, respectively), indicating the capacity to neutralize the virus is most superior by infection alone. Conclusions: Vaccines induce anti-RBD titers as high as the natural infection with lower neutralization capacity, and it does not boost immunity in pre-infected persons.

## 1. Introduction

The coronavirus disease 2019 (COVID-19) pandemic started in late 2019 when the severe acute respiratory syndrome virus (SARS-CoV-2) quickly spread from China, posing a health crisis globally [1,2,3]. As countries grappled with the increasing number of infections, healthcare systems faced unprecedented challenges [4]. Hospitals overflowed, resources were stretched thin, the global economy took a massive hit as businesses closed, and travel came to a near halt [5]. The pandemic shed light on the importance of swift, coordinated global responses and the necessity for science-driven strategies to combat future health crises [6]. 

The SARS-CoV-2 virus mainly affects the respiratory system, resulting in a broad range of clinical symptoms from mild ones, such as flu-like sickness, to severe ones, such as breathing difficulties, and even death due to sepsis, acute cardiac damage, heart failure, and multi-organ dysfunction [7,8]. Thus, individual immunity against the virus contributing to herd immunity is crucial. 

The genome of the virus encodes four structural proteins: the spike (S), nucleocapsid (N), membrane (M), and envelope (E) proteins, analogous to SARS-CoV-1 [1,7,9,10]. The S-protein, particularly its receptor-binding domain (RBD), is essential for infecting host cells by starting cell penetration when it attaches to the angiotensin-converting enzyme 2 (ACE2) receptor. The S protein is a trimeric glycoprotein and belongs to class I fusion proteins containing two subunits, Spike 1 and Spike 2, mediating attachment and fusion of viral and cellular membranes. Other structural proteins form the ribonucleoprotein core that drives viral assembly [1,11,12,13]. Grasping the subtleties of these proteins has been crucial for creating reliable diagnostic methods, therapies, and immunizations [14]. Given the virus’s mutability, the S-protein has especially drawn considerable global research interest. This emphasis is to predict shifts that might affect the effectiveness of vaccines and influence the virus’s spread and severity [15]. 

Natural infection elicits adaptive immune responses against the structural proteins, with T cells and antibodies being the key components [16]. Many studies have reported that SARS-CoV-2 patients exhibited T-cell immunity and neutralizing antibodies [17,18,19,20,21]. The antibody frequently targets the RBD of the spike protein to prevent the virus from interacting with the ACE2 receptor and starting a productive infection [16,20,22]. Neutralizing antibodies are probably an essential correlate of COVID-19 protection [22,23,24,25] and are associated with protective immunity against second infection [26].

In a large population, herd immunity can be achieved either by vaccination or natural infection [16]. Several types of vaccines have been produced, including the inactivated SARS-CoV-2 vaccine, which was used in 40 countries [27]. The levels of neutralizing antibodies produced by the SARS-CoV-2 vaccines have been compared to those in naturally infected people [22,28]. It raises the question of whether or not vaccines are needed to boost immunity in people with past infections. It was suggested that hybrid immunity will be developed by vaccination in people with pre-existing immunity [29]. Neutralization titers in vaccinated individuals were markedly higher than those in unvaccinated individuals with pre-infection across several variants of SARS-CoV-2 [30]. Several studies also revealed that neutralizing antibody titer induced by vaccination was higher among people with past infection than naive individuals [31,32,33]. It was suggested that one dose of inactivated vaccines or mRNA vaccines is enough for subjects with pre-existing immunity to boost the antibody titer [31,32,34,35]. On the contrary, a study reported that infection-acquired immunity was higher in unvaccinated individuals than subsequently vaccinated individuals [36].

The race to develop a vaccine saw an unprecedented level of international collaboration among scientists, researchers, and pharmaceutical companies [37]. However, with vaccine development came challenges like distribution, equity in access, and public hesitancy [38]. As more data emerge about the interplay between natural immunity and vaccination, global health strategies evolve to prioritize populations most in need and to ensure maximum protection [39].

Therefore, it is essential to clarify whether the vaccination is needed for people with previous infections to save the cost and the burden of vaccine production, especially in Indonesia, where no domestic vaccine has yet been distributed. Indonesia’s unique demographic and geographical challenges require tailored strategies. The results of this study will not only inform Indonesia’s pandemic response but could also provide valuable insights for other countries with similar challenges [40]. As global entities look for efficient ways to curb the pandemic, comprehensive studies like these play a crucial role in guiding informed decisions, optimizing resources, and ultimately safeguarding public health [41]. Thus, this study aimed to compare the antibody titer between vaccinated individuals with and without pre-existing immunity and unvaccinated convalescent individuals. In addition, it evaluated the kinetics of virus-neutralizing antibodies in a prospective cohort. 

## 2. Materials and Methods

A prospective cohort involving COVID-19-recovered individuals and vaccine recipients visiting Dr. Tadjuddin Chalid Hospital and Dr. Wahidin Soedirohusodo Hospital was conducted in Makassar, the capital city of South Sulawesi province in Indonesia. All eligible subjects were selected by purposive sampling from April 2021 to December 2021. We recruited vaccine recipients with and without an infection history and unvaccinated survivors. The inclusion criteria were COVID-19 survivors or COVID-19 vaccine recipients aged above 17 years old due to age restriction for COVID-19 vaccine at that time, having completed two doses of the inactivated whole SARS-CoV-2 virus vaccine CoronaVac^®^ from Sinovac at a 4-week interval, or being confirmed recovered from COVID-19 by having a converted swab PCR test result from positive to negative (for unvaccinated subjects), and willing to participate until day 90 by signing written informed consents. Blood was withdrawn at days 0, 30, and 90 after the negative PCR result or after the second dose of immunization. At the end of the study, since several COVID-19 infections occurred after vaccination, we also recruited individuals with such breakthrough infections. Still, their blood was withdrawn at day 0 only due to the limit of the study time frame. This study was approved by the Institutional Ethics Committee of the Faculty of Medicine, Hasanuddin University (753/UN4.6.4.5.31/PP36/2022). Informed consent was obtained from all patients enrolled in this study.

Sera were separated from median cubital vein blood samples collected from volunteers. After disinfection of the venipuncture side with a 70% alcohol swab, whole blood was collected from the superficial vein of the upper limb using a butterfly needle (BD Vacutainer collection set). The blood was flowed into the serum separator tube (BD VacutainerTB Venous Blood Collection Tube: SSTTM Serum Separation Tube Hemograd) up to 5–7 mL. Collected blood was centrifuged at the Hasanuddin University Medical Research Centre (HUMRC) of Hasanuddin University Hospital for serum separation. Then, sera were aliquoted 100 µL into each 0.5 µL Eppendorf tube to avoid repeated freeze-thawing. All sera were kept at −80 °C before being subjected to any experiments. 

Indirect ELISAs were performed using the commercial human embryonic kidney (HEK293) HPLC-verified (Sino Biological, Beijing, China #40591-V08H) RBD protein as the antigens. The 96-well microplates (Costar EIA/RIA polystyrene plates, Corning Inc., Corning, New York, NY, USA #3590) were coated with 0.2 μg/mL of the antigen dissolved in phosphate-buffered saline (PBS), pH 7.4 per well, and incubated overnight at 4 °C. The plates were blocked with 1% bovine serum albumin (BSA) in PBS (pH 7.4), washed with PBS-T, and incubated with serum samples diluted 1:100 in PBS containing 1% BSA. After being washed with PBST three times and with PBS once, the plates were incubated with a secondary antibody, horseradish peroxidase (HRP)-conjugated anti-human IgG (Bio-Rad Lab, Inc., Tokyo Japan) recognizing an Fc domain of human IgG. After incubation, the plates were washed, then 100 μL/well of peroxidase substrate solution (H_2_O_2_ and 2,2′-azino-bis(3-ethylbenzothiazoline-6-sulfonate)) was added to each well, and the plates were incubated for 30 min at room temperature for color development as described elsewhere. The endpoint titer was expressed as the absorbance at 414 nm on a microplate reader. For this experiment, we also used 30 samples from the pre-pandemic era as negative controls.

Neutralizing activity of the serum was examined using a VSV-based pseudovirus, as previously described [42]. Briefly, the pseudovirus was engineered to express the Wuhan-Hu-1 SARS-CoV-2 Spike protein on the viral surface, in which the luciferase gene was incorporated into the viral genome. The serum was diluted with the medium, and the virus was added in triplicate. The final dilution rate of the serum was 1:100. The mixture of the virus and the serum was incubated with human embryonic kidney (HEK) 293T cells that expressed human ACE2 and human TMPRSS2. The cells were examined by Luciferase Assay System (Promega, Madison, WI, USA) after 24 h of incubation.

Statistical analysis was performed using GraphPad Prism version 9.0 for Mac OS. The Kolmogorov–Smirnov test was used to analyze the normality of the distribution of the antibody titers. One-way ANOVA followed by Tukey’s multiple comparison test was used for normally distributed data, whereas the Kruskal–Wallis test with a post hoc Dunn’s multiple comparison test was used to compare differences in antibody titers between groups. A *p*-value of <0.05 was considered statistically significant. Spearman’s test was used to analyze the correlations between optical densities (ODs) and the percentage of internalization.

## 3. Results

### 3.1. Antibody Titer Elicited by Vaccination and Natural Infection

We collected 356 samples from 156 subjects at baseline (Table 1), consisting of 35 unvaccinated survivors (Group 1), 52 vaccinated persons with a pre-infection (Group 2), 41 naïve vaccine recipients (Group 3), and 28 persons with breakthrough infections (Group 4). The study was completed on day 90 by 90 subjects consisting of 14 COVID-19 unvaccinated survivors, 41 vaccine recipients with pre-infection, and 35 naïve vaccine recipients. 

The characteristics of the study subjects are shown in Table 2. The median age of the four groups was 31–48 years old (range 18–72). The number of severe and non-severe cases in the unvaccinated survivor group was comparable. However, in the vaccinated survivors, all cases were categorized as non-severe. The majority of subjects in groups 1 and 2 had normal body mass index (BMI). In contrast, greater number of individuals were underweight in groups 3 and 4. The median duration between infection and first dose vaccination in group 2 was four months (range 3–11 months). On the other hand, breakthrough infections occurred 26 months post-second vaccine dose (median four months) among group 4.

Analysis of anti-RBD antibody titers between groups showed a significant difference on day 0 between all groups to pre-pandemic sera (Figure 1A; *p* < 0.0001). In addition, the antibody titer of persons with breakthrough infections was significantly lower than that of vaccinated naïve individuals (*p* = 0.0065), whereas no significant differences were observed among other seropositive groups. In further detail, the RBD level of vaccinated survivors was similar to that of the unvaccinated ones. On the other hand, the level of this antibody in vaccinated individuals with and without pre-existing immunity was identical.

Between-group analysis on samples collected on day 30 and day 90 also demonstrated that the antibody level was not significantly different (Figure 1B; *p* = 0.2535 and Figure 1C; *p* = 0.6249, respectively). 

Within-group analysis was conducted for each group to compare antibody titers between days 0, 30, and 90. In groups 1 and 3, there were no significant discrepancies in the antibody levels on days 0, 30, and 90 (*p* = 0.9691 and *p* = 0.4004, respectively). In group 2, the antibody levels on days 0, 30, and 90 were distinctive (*p* = 0.0027) since there was an increase in the antibody level on day 30 compared with that on day 0. However, the antibody level on day 90 was not distinct significantly from that of day 0 and day 30, indicating that there was a decrease in the antibody to a comparable level as day 0, but this was insignificant compared with the level on day 30.

### 3.2. The Neutralization Capacity of the Antibody

After determining the antibody titers of all samples, we randomly selected 36 samples for groups 1, 2, and 3 and 28 samples for group 4 to analyze the capacity of the antibodies to neutralize the viruses. The sera were mixed with the Wuhan-Hu-1 pseudovirus, and then the percentage of virus internalization to the cells, compared with non-serum control by 100%, was calculated by the luciferase activity. The result revealed that the antibodies of unvaccinated survivors neutralized the virus more superiorly (Figure 2A, Group 1; percentage of internalization 37.26 ± 37.88) than those of vaccinated individuals without pre-infection (Group 3; 48.15 ± 45.13). The mean percentage of viral internalization of the vaccinated persons with a history of infection and the infected person after having two doses of vaccine were 37.02 ± 43.13 (Group 2) and 62.95 ± 41.28 (Group 4), respectively. There were no statistical differences among all groups. 

We then analyzed the correlation of antibody titers with the virus neutralization using Spearman’s test (Figure 2B). We observe a strong correlation between the reduction in viral internalization and the antibody titer in unvaccinated survivors (*r* = −0.8558; 95% Confidence Interval (CI), −0.9259 to −0.7288; *p* < 0.0001). The antibodies of vaccinated individuals without pre-infection showed the weakest correlation with the reduction of viral internalization (*r* = −0.581; 95% CI, −0.7679 to −0.3028; *p* = 0.0002). The correlation coefficient of the vaccinated persons with a history of infection and the infected person after having two doses of vaccine were −0.7855 (95% CI, −0.8877 to −0.6096) and −0.6889 (95% CI, −0.8481 to −0.4156). 

Further analysis was conducted based on the sample collection time (Figure 3). The natural antibodies could consistently neutralize the virus until day 90 (Group 1, Figure 3A–C). The vaccinated naïve individuals also show consistent neutralization capacity until day 90, although at a lower level (Group 3, Figure 3A–C). The highest neutralization capacity was shown on day 30 after the second dose in the vaccinated persons with pre-infection. However, it decreased to a lower level than the vaccinated persons without pre-infection on day 90 (Group 2, Figure 3A–C). In group 4, the proportion of the anti-RBD with high levels was low (Figure 1A), and thus, the neutralization activity was relatively low in total (Figure 3A).

## 4. Discussion

In this study, we investigated the antibody induction against SARS-CoV-2 by natural infection and whole-inactivated vaccine. We involved various groups according to the history of infection and vaccination of the study subjects to obtain more comprehensive data. From this study, we could reveal some information regarding the coronavirus-targeting antibody. First, it shows whether the antibody level induced by vaccination is higher than that induced by natural infection. In addition, it investigates whether vaccination to people with pre-existing immunity boosts the antibody titer. Moreover, it displays the dynamics of the antibody titer up to 90 days. Finally, it describes the neutralization capacity of the antibodies induced by natural infection, vaccine, or the combination of both against the original strain of SARS-CoV-2. 

In this study, the whole-inactivated vaccines induced RBD antibodies at the same level as the natural infection 30 days after the first dose. This result is contradictive with a study reporting that the mRNA-1273 vaccine induced a higher titer of neutralizing antibodies than natural infection at around 100-fold higher than the natural infection alone [29,43]. Another study comparing the antibody level of 41 convalescent sera and 28 mRNA vaccine sera—Pfizer or moderna—also found that the level of RBD antibody was 17-fold higher in the vaccine sera than in the convalescent sera [44]. 

Our data also revealed that the antibody titers of survivors were not boosted by vaccination since there was no difference between the antibody titer of vaccinated survivors compared with vaccinated naïve individuals and the unvaccinated survivors on days 0, 30, and 90 after the second dose injection. The average time of receiving the first vaccine from days of infection was four months, when the antibody was much more likely to be at a high level, since a study suggested that neutralizing antibodies peaked at 120 days after onset and are still detectable for over one year [27]. The second dose increased the titer after 30 days of injection in pre-infected individuals but was still the same with the titer of antibodies induced by the vaccine alone. Thus, the expected hybrid immunity was not elicited by the whole inactivated vaccination to pre-infected persons, contrary to a study reporting that vaccination with mRNA-1273 increased the neutralizing antibodies 25 times higher in the pre-infected persons compared with vaccinated persons without pre-infection [29]. However, our result was in agreement with a study on 35,768 healthcare workers in the UK, which found that the humoral responses of unvaccinated survivors remained consistently higher than those who received two doses of BNT162b2 [36]. Therefore, the induction of antibody titer among individuals with pre-existing immunity might depend on the vaccine type. We assume the inactivated vaccines might not be as potent as mRNA-1273 vaccines in inducing neutralizing antibodies in pre-infected people.

The neutralization capacity of the RBD antibodies elicited by COVID-19 natural infection to those induced by inactivated vaccines of the original strain of SARS-CoV-2 was conducted by analyzing the correlation of the anti-RBD titers with the viral cell internalization. The neutralizing epitopes on the RBD of the spike protein are highly immunogenic, as was primarily for the domain that binds with the ACE2 receptor, so it was suggested that a single mutation could not avoid human polyclonal antibody neutralization [26]. Our data demonstrated that the neutralization capacity of RBD antibodies elicited by natural infection is better than that induced by whole-inactivated vaccine. Vaccination in pre-infected individuals increases the capacity to neutralize the virus 30 days after completion, but it declined after two months.

Since its emergence, SARS-CoV-2 has undergone mutations causing variants of concern (VoCs) and variants of interests (VoIs), some of which are highly transmissible and capable of escaping antibody neutralization from natural infection or vaccines [26,29,45,46,47,48]. Several mutations have changed the RBD conformation that may disturb antigen recognition [47]. In the current study, we did not analyze the neutralization capacity of the variants of SARS-CoV-2. A study reported that the mRNA-vaccinated sera neutralizes the variants of SARS-CoV-2 more effectively than convalescent sera [44], including the N501Y variant. Therefore, future studies are needed to evaluate the neutralization capacity of the whole-inactivated vaccine sera to the new variants of SARS-CoV-2. Since many individuals have taken a booster dose using the mRNA vaccines, a future study on its effect on viral neutralization may also be essential.

In this study, we found that several samples from the pre-pandemic era show high anti-RBD titers. Similar results have been reported by other studies, which suggested that the positivity shown by the samples from the pre-pandemic era might be due to cross-reactivity by previous infection with other coronaviruses, such as common cold-causing coronavirus [49,50]. 

In our study, several samples of breakthrough infection showed antibodies at a matching level to samples from the pre-pandemic era. Low antibody titers correlate strongly with low capacity to neutralize the virus. Since there were no baseline data for group 4, we assumed the breakthrough infection occurred among vaccinated people with low antibody titers (non-responders). Thus, we concluded that vaccines are essential for those without pre-existing immunity to prevent them from contracting COVID-19. 

Finally, there are several limitations of this study: (1) it merely investigated the neutralization activity of the original Wuhan-Hu strain, (2) there were no baseline data for both the infection and vaccine groups; we assumed that the antibodies were at the expected level, (3) we did not analyze the variables of the subjects, such as age, gender, disease severity, body mass index, and exposure-related work that may affect the antibody response and their neutralization capacity, as conducted by a prior study by Kodde et al., due to limited sample size [51,52]. It was reported that an increase in age decreases the neutralizing antibody [51,53]. Previous studies demonstrated a relationship between BMI and COVID-19 fatality [54]; subjects with a BMI of <18.5 kg/m^2^ and those with a BMI ≥ 25 kg/m^2^ had a high risk of fatal illness. A more severe disease also correlates with a higher neutralizing antibody [21].

However, despite the limitations, the strengths of our study are as follows: (1) it has a comprehensive analysis of the antibody dynamic with a prospective cohort approach until 90 days of follow-up, and (2) it includes several groups with diverse courses of either COVID-19 vaccination or infection. To our knowledge, this is the first study in Indonesia showing the divergence of neutralizing antibodies induced by vaccines and infection. These data are essential because recently, the WHO recommended that the Indonesian government incorporate COVID-19 vaccinations into routine services despite the excellent coverage of the vaccination campaigns due to the waning interest in booster doses [55]. If so, to decrease the demand for vaccines and to save resources, prioritizing persons without a history of natural infection should be considered, as our data suggest that vaccination did not produce better neutralizing antibodies in persons with a history of infection, yet it may be essential in preventing diseases among naïve individuals. In addition, research in the development of whole-inactivated vaccines should be conducted to improve the neutralization capacity of the elicited antibodies, especially against new variants of SARS-CoV-2. 

## 5. Conclusions

This study revealed that vaccination with whole-inactivated vaccines does not increase the antibody titer against RBD in individuals with pre-existing immunity. Moreover, the viral neutralization capacity of vaccine serum is relatively low compared with that of the convalescent serum. However, the vaccine may be crucial in preventing the disease among people without an infection history.

## Figures and Tables

**Figure 1 antibodies-12-00060-f001:**
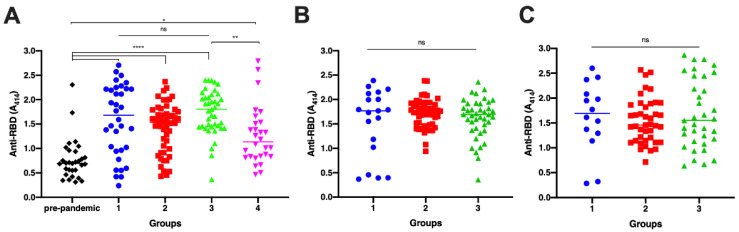
Comparison of anti-RBD IgG antibody titers among all groups. Pre-pandemic = sera from Sumba Island collected in the pre-pandemic era (black diamond); Group 1 = unvaccinated survivors; Group 2 = vaccinated persons with pre-infection; Group 3 = vaccinated persons without pre-infection; and Group 4 = persons with breakthrough infection. Blood was withdrawn on days 0, 30, and 90 post-infection for group 1. Sera was withdrawn on days 0, 30, and 90 post-second doses of whole inactivated vaccine for group 2 and group 3. ELISA was performed to measure the antibody titer for samples collected on day 0 (**A**), day 30 (**B**), and day 90 (**C**). Individual data points are shown with the median (midline). The difference between groups was analyzed by Kruskal–Wallis (**A**) and ANOVA (**B**,**C**). ns, non-significant; * *p* < 0.05, ** *p* < 0.01; **** *p* < 0.0001.

**Figure 2 antibodies-12-00060-f002:**
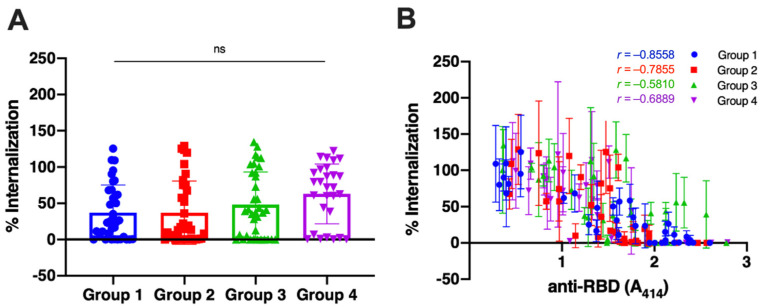
Comparison of viral neutralization of sera samples among all groups. Group 1 = unvaccinated survivors; Group 2 = vaccinated persons with pre-infection; Group 3 = vaccinated persons without pre-infection; and Group 4 = persons with breakthrough infection. (**A**) Neutralization of the Wuhan-Hu-1 pseudovirus by the serum samples as determined by luciferase assay. Each symbol represents an individual means of viral internalization from triplicate data, and the bar represents the mean with the standard deviation of the group. The difference between groups was analyzed by a Kruskal–Wallis test. ns: not significant. (**B**) The percentage of internalization was plotted against the titer of each sample. The Spearman correlation coefficient between the two variables was calculated for each group. The line and error bars represent individual data’s mean and standard deviation.

**Figure 3 antibodies-12-00060-f003:**
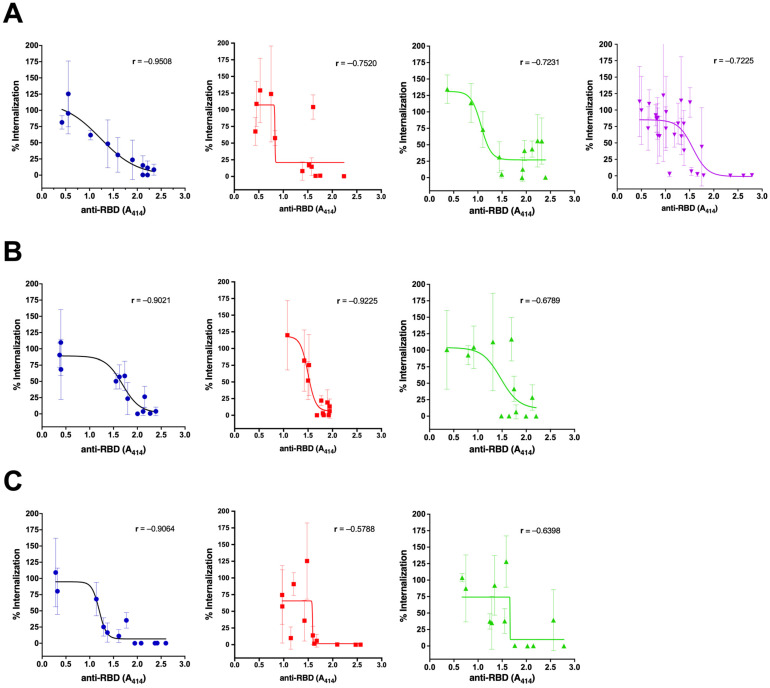
Comparison of viral neutralization of sera samples among all groups based on anti-RBD antibody titer on day 0 (**A**), day 30 (**B**), and day 90 (**C**). Group 1 = unvaccinated survivors (blue dots); Group 2 = vaccinated persons with pre-infection (red squares); Group 3 = vaccinated persons without pre-infection (green triangles); and Group 4 = persons with breakthrough infection (purple triangles). Samples from experiments in Figure 1 were selected to represent each group’s low, medium, and high titers for every collection time. The line and error bars represent the mean and standard deviation. The Spearman correlation coefficient between the antibody titer and the percentage of viral internalization was calculated for each group.

**Table 1 antibodies-12-00060-t001:** Status of the recipients in each group.

Group	COVID-19 Infection	Vaccination	n
1	Yes	No	35
2	Yes	Yes (After infection)	52
3	No	Yes	41
4	Yes	Yes (Before infection)	28

**Table 2 antibodies-12-00060-t002:** Characteristics of study subjects.

Variable	Group 1	Group 2	Group 3	Group4
Age				
Mean	46.7	36.6	34.4	40.1
Median (Range)	48 (19–72)	36.0 (24–56)	31.0 (18–70)	41.0 (23–64)
Sex				
Female (%)	19 (54.3)	18 (34.6)	19 (46.3)	11 (39.3)
Male (%)	16 (45.7)	34 (65.4)	22 (53.7)	17 (60.7)
Severity				
Severe (%)	17 (48.6)	0 (0)	N/A	1 (3.6)
Non-Severe (%)	18 (51.4)	52 (100)	N/A	27 (96.4)
Body Mass Index (WHO)				
Underweight (%)	11 (31.4)	12 (23.1)	23 (56.1)	11 (39.3)
Normal (%)	21 (60.0)	26 (50.0)	16 (39.0)	8 (28.6)
Overweight (%)	3 (8.6)	13 (25.0)	2 (4.9)	9 (32.1)
Obese (%)	0 (0)	1 (1.9)	0 (0)	0 (0)
Health Workers				
Yes (%)	1 (2.9)	32 (61.5)	0 (0)	3 (10.7)
No (%)	34 (97.1)	20 (38.5)	41 (100)	25 (89.3)
Duration between infection and vaccine (months)				
Mean	N/A	4.70	N/A	4.034
Median (Range)	N/A	4 (3–11)	N/A	4 (2–6)

## Data Availability

The raw data presented in this study will be made available to any researchers on request to the corresponding author.

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
