# Peer review of "Comparison between Neutralization Capacity of Antibodies Elicited by COVID-19 Natural Infection and Vaccination in Indonesia: A Prospective Cohort"

_2073-4468, 2023, doi:10.3390/antib12030060_

Round 1
Reviewer 1 Report
Thank you for the opportunity to review this article. The topic is very relevant since it addresses an issue that may lead to changes in policy which may lead to cost-effective strategies in the prevention of COVID-19. The discrepancies in the previous studies also make this study relevant. I have some major and minor revisions required below.
MAJOR REVISIONS
Materials and Methods
1. No reason is given as to why only participants above the age of 17 years were recruited. Could it be that they were the only ones eligible for vaccination at the time? If so, add this to the manuscript.
2. Although the ethical clearance is added at the end of the manuscript, it should also be stated under this section that it was sought.
3. It is not explained in the manuscript how the sample size was determined. This should be added to this section.
4. It is not explained how the sampling of participants was performed. Readers will be interested in this kind of information.
Results
When presenting results, authors mix the results with discussion in some sections. This section should just present results. Any explanations should be removed and added to the discussion section. For example:
5. ‘These data inferred the antibody titers of survivors were not boosted by vaccination on day 28 after first dose-injection. On the other hand, breakthrough infection did not increase the antibody titer elicited by vaccines. Since no baseline data for group 4, we assumed the breakthrouh infection occured among vaccinated people with low antibody titer (non-responders).’ This should go under discussion.
6. ‘These results suggest that natural infection alone induced more effective antibodies than whole-inactivated-vaccine and no booster effect from vaccines to the existing immunity.’ This should go under discussion.
7. ‘All data suggest that vaccination with whole-inactivated vaccine does not effectively induce antibody titer with a strong neutralization capacity for persons with history of COVID-19 natural infection.’ This should go under discussion.
Discussion
8. I suggest that all the limitations are presented at the end of the discussion.
9. Add strengths of your study before the limitations are presented. There are some already in the discussion.
Conclusion
10. There should be a stand-alone conclusion section for the manuscript.
MINOR REVISIONS
There are a number of grammatical and spelling errors that will need to be corrected.
11. In the second paragraph of introduction, you state, ‘Natural infection indeed elicited adaptive immune responses to the virus’ structural proteins, with T cells and antibodies being the key components.’ Remove ‘indeed’ and ‘elicited’ should be changed to ‘elicit’.
12. In the third paragraph of introduction, you state, ‘In a large population, herd immunity can be achieved by vaccination besides the natural infection’. Put a comma after vaccination.
13. In the third paragraph of introduction, you state, ‘This raises the question of whether or not vaccines is needed to boost immunity in people with past infections.’ Change ‘is’ to ‘are’.
14. In the fourth paragraph of introduction, you state, ‘The result showed that vaccination does not increase the antibody titer against RBD in individuals with pre-exisiting immunity. Moreover, the capacity of viral neutralization of vaccine serum is not as good as that of the convalescent serum.’ Remove this. You cannot present your results in the introduction.
15. In the last paragraph of materials and methods, you state, ‘One-way ANOVA followed by Tukey’s multiple comparison test was used for normally distributed data,…’ Remove ‘was’ and put ‘were’.
16. In the last paragraph of materials and methods, you state, ‘…whereas Kruskal-Wallis test with a post hoc Dunn’s multiple comparison test was used to compare differences in antibody titers between groups.’ Replace ‘was’ with ‘were’.
Results
17. In section 3.1, you state, ‘The characteristics of the study subjects were shown in table 2.’ Replace ‘were’ with ‘are’.
18. Give a summary of the important findings in Table 2 above the table.
19. In section 3.2, you state, ‘No statistical differences among all groups.’ Write a full sentence.
Discussion
20. In the first paragraph of discussion, you state, ‘In this study, we compare the capacity of the RBD antibodies elicited by COVID-19….’ Replace ‘compare’ with ‘compared’.
21. In the second paragraph of discussion, you state, ‘Since its emergence, SARS-CoV-2 has undergone mutations causing variants of concern and variants of interest, some highly transmissible and are capable of escaping the antibody neutralization either from natural infection or vaccines.’ Remove ‘are’.
22. In the second paragraph of discussion, you state, ‘Nonetheless, since the whole-inactivated vaccines generated from the original strain as well,…’ Add ‘are’ between ‘vaccines’ and ‘generated’.
23. In the second paragraph of discussion, you state, ‘We predict that even the inactivated vaccines is generated using the later strains of SARSCoV-2, …’ Replace ‘is’ with ‘are’.
Minor revisions required.
Author Response
Thank you very much for your suggestions which enabled us to greatly improved the quality of our manuscript. In the following pages are our point-by-point responses. We have revised the manuscript according to all points addressed. Revisions in the text are shown using track changes. We also attach the clear version of the revised manuscript. We hope the revisions in the manuscript and our accompanying responses will be sufficient to make our manuscript suitable for publication in antibodies.
Reviewer 1
MAJOR REVISIONS
Materials and Methods
- No reason is given as to why only participants above the age of 17 years were recruited. Could it be that they were the only ones eligible for vaccination at the time? If so, add this to the manuscript.
COVID-19 survivors were age above 17 years old because at the time, they were only the individuals who qualified for vaccination.
- Although the ethical clearance is added at the end of the manuscript, it should also be stated under this section that it was sought.
We have added “This study was approved by the Institutional Ethics Committee of the Faculty of Medicine, Hasanuddin University (753/UN4.6.4.5.31/PP36/2022). Informed consent was obtained from all patients enrolled in this study” in the methods.
- It is not explained in the manuscript how the sample size was determined. This should be added to this section.
As we use purposive sampling, sample size was not determined. We analyzed all eligible samples that could be recruited within the study time-frame.
- It is not explained how the sampling of participants was performed. Readers will be interested in this kind of information
We have added in the methods about purposive sampling.
Results
- ‘These data inferred the antibody titers of survivors were not boosted by vaccination on day 28 after first dose-injection. On the other hand, breakthrough infection did not increase the antibody titer elicited by vaccines. Since no baseline data for group 4, we assumed the breakthrouh infection occured among vaccinated people with low antibody titer (non-responders).’ This should go under discussion.
- ‘These results suggest that natural infection alone induced more effective antibodies than whole-inactivated-vaccine and no booster effect from vaccines to the existing immunity.’ This should go under discussion.
- ‘All data suggest that vaccination with whole-inactivated vaccine does not effectively induce antibody titer with a strong neutralization capacity for persons with history of COVID-19 natural infection.’ This should go under discussion.
Thank you for the suggestions. All those points have been addressed and moved to the discussion.
Discussion
- I suggest that all the limitations are presented at the end of the discussion.
We have moved all the limitations at the end of the discussion, before our closing paragraph.
- Add strengths of your study before the limitations are presented. There are some already in the discussion.
This point has been added in the closing paragraph after limitation because it is more related to our points addressed in the closing.
Conclusion
- There should be a stand-alone conclusion section for the manuscript.
This item has been added at the end of the manuscript.
MINOR REVISIONS
There are a number of grammatical and spelling errors that will need to be corrected.
- In the second paragraph of introduction, you state, ‘Natural infection indeed elicited adaptive immune responses to the virus’ structural proteins, with T cells and antibodies being the key components.’ Remove ‘indeed’ and ‘elicited’ should be changed to ‘elicit’.
- In the third paragraph of introduction, you state, ‘In a large population, herd immunity can be achieved by vaccination besides the natural infection’. Put a comma after vaccination.
- In the third paragraph of introduction, you state, ‘This raises the question of whether or not vaccines is needed to boost immunity in people with past infections.’ Change ‘is’ to ‘are’.
- In the fourth paragraph of introduction, you state, ‘The result showed that vaccination does not increase the antibody titer against RBD in individuals with pre-exisiting immunity. Moreover, the capacity of viral neutralization of vaccine serum is not as good as that of the convalescent serum.’ Remove this. You cannot present your results in the introduction.
- In the last paragraph of materials and methods, you state, ‘One-way ANOVA followed by Tukey’s multiple comparison test was used for normally distributed data,…’ Remove ‘was’ and put ‘were’.
- In the last paragraph of materials and methods, you state, ‘…whereas Kruskal-Wallis test with a post hoc Dunn’s multiple comparison test was used to compare differences in antibody titers between groups.’ Replace ‘was’ with ‘were’.
All those points have been corrected accordingly.
Results
- In section 3.1, you state, ‘The characteristics of the study subjects were shown in table 2.’ Replace ‘were’ with ‘are’.
This point has been corrected accordingly.
- Give a summary of the important findings in Table 2 above the table.
We have added summary of the table 2 in the result section.
- In section 3.2, you state, ‘No statistical differences among all groups.’ Write a full sentence.
We have changed the sentence accordingly.
Discussion
- In the first paragraph of discussion, you state, ‘In this study, we compare the capacity of the RBD antibodies elicited by COVID-19….’ Replace ‘compare’ with ‘compared’.
- In the second paragraph of discussion, you state, ‘Since its emergence, SARS-CoV-2 has undergone mutations causing variants of concern and variants of interest, some highly transmissible and are capable of escaping the antibody neutralization either from natural infection or vaccines.’ Remove ‘are’.
- In the second paragraph of discussion, you state, ‘Nonetheless, since the whole-inactivated vaccines generated from the original strain as well,…’ Add ‘are’ between ‘vaccines’ and ‘generated’.
- In the second paragraph of discussion, you state, ‘We predict that even the inactivated vaccines is generated using the later strains of SARSCoV-2, …’ Replace ‘is’ with ‘are’.
All those points have been corrected accordingly

Reviewer 2 Report
This is an interesting manuscript by Sitti Nurisyah et.al. “Comparison between neutralization capacity of antibodies elicited by COVID-19 natural infection and vaccination in Indonesia: A prospective cohort” demonstrating the anti-RBD IgG titers and their neutralizing capacity of pseudovirus expressing express the Wuhan-Hu-1 SARS-CoV-2 Spike protein among COVID-19 vaccinated and non-vaccinated with or without COVID-19 infection. Overall, the methodologies and assays chosen are sound and the conclusions made are well-founded with the presented results. However, there are few places in the manuscript in which the presentation must be improved as noted below.
1. There was a difference in the number of study recipients between table 1 and table 2. Group 1 in table 1 showed n=32, but the number was not matching in table 2 characteristics like severity, BMI, sex, health workers. Similar with group 4.
2. Why there was an anti-RBD titers in pre-pandemic population. The levels in 2 individuals are more than mean value in group 1-3.
Author Response
Thank you very much for your suggestions which enabled us to greatly improved the quality of our manuscript. In the following pages are our point-by-point responses. We have revised the manuscript according to all points addressed. Revisions in the text are shown using track changes. We also attach the clear version of the revised manuscript. We hope the revisions in the manuscript and our accompanying responses will be sufficient to make our manuscript suitable for publication in antibodies.
- There was a difference in the number of study recipients between table 1 and table 2. Group 1 in table 1 showed n=32, but the number was not matching in table 2 characteristics like severity, BMI, sex, health workers. Similar with group 4.
We have revised the tables after re-checking the data.
- Why there was an anti-RBD titers in pre-pandemic population. The levels in 2 individuals are more than mean value in group 1-3.
The antibodies shown by the pre-pandemic samples might be due to cross reactivity with previous infection by other coronavirus. It has been shown by another study and we have addressed this point in the discussion.

Reviewer 3 Report
To Author:
In this article, the authors compared the antibody titers and neutralizing capacity induced by inactivated vaccines and COVID-19 natural infection. The authors found that vaccination does not increase the antibody titers against RBD (receptor-binding domain) in individuals with pre-existing immunity. I considered this research article to be significant. However, I have several suggestions.
Comments:
(1) I think the number of patients analyzed by the authors in this article is not large enough and the authors should increase the number of patients if possible.
(2) COVID-19 affects people of different ages differently, and the authors should consider patient age as an important factor when analyzing the results.
(3) The authors’ discussion in the article is too simplistic, and a section should be added to discuss the impact of different vaccines on infection with different COVID-19 mutant strains.
Author Response
Thank you very much for your suggestions which enabled us to greatly improved the quality of our manuscript. In the following pages are our point-by-point responses. We have revised the manuscript according to all points addressed. Revisions in the text are shown using track changes. We also attach the clear version of the revised manuscript. We hope the revisions in the manuscript and our accompanying responses will be sufficient to make our manuscript suitable for publication in antibodies.
(1) I think the number of patients analyzed by the authors in this article is not large enough and the authors should increase the number of patients if possible.
We used purposive sampling, and we recruited all eligible subjects within the study time-frame. It is not possible to increase the number of samples, because the mutant strain (e.g. Omicron) has already been spreading in the community of Indonesia, and currently we cannot define people infected with the early Wuhan strain.
(2) COVID-19 affects people of different ages differently, and the authors should consider patient age as an important factor when analyzing the results.
We have put this into the limitation. We did not analyze the effect of these factors due to low sample size.
(3) The authors’ discussion in the article is too simplistic, and a section should be added to discuss the impact of different vaccines on infection with different COVID-19 mutant strain
In this study, we only used 1 type of vaccine CoronaVac® from Sinovac. However, we have addressed that future studies are needed for evaluating different strains of SARS-CoV-2.

Round 2
Reviewer 1 Report
Thank you for addressing all my comments.
Reviewer 2 Report
Dear Authors,
Thank you for addressing the comments.
Reviewer 3 Report
The revised paper is ok. I do not have any comments.